# Pembrolizumab-Induced Lichen Planus Pemphigoides in a Patient with Metastatic Adrenocortical Cancer: A Case Report and Literature Review

Vrinda Madan, Mark C. Marchitto and Joel C. Sunshine *

Department of Dermatology, Johns Hopkins University School of Medicine, Baltimore, MD 21287, USA; vmadan2@jhmi.edu (V.M.)
* Correspondence: joelsunshine@jhmi.edu

**Abstract:** While the advent of immune-checkpoint inhibitors has revolutionized cancer therapy, immune-related adverse effects (irAEs) have also been on the rise. Cutaneous toxicities are among the most common irAEs, especially in the context of programmed cell death protein-1 (PD-1) inhibitors like pembrolizumab. Herein, we report a case of anti-PD-1-induced lichen planus pemphigoides (LPP)—a rare autoimmune blistering disorder with characteristics of both lichen planus and bullous pemphigoid. To our knowledge, this is the first reported case of LPP following anti-PD-1 therapy for metastatic adrenocortical cancer. Recognizing that LPP is within the spectrum of irAEs is important, especially as the indications for immunotherapy grow to include rarer malignancies like adrenocortical cancer. In addition to our case presentation, we also provide a comprehensive review of the literature surrounding immunotherapy-induced LPP—highlighting key characteristics towards the early recognition and clinical management of this cutaneous irAE.

**Keywords:** lichen planus pemphigoides; pembrolizumab; anti-PD-1; immunotherapy; immune-related adverse effects; adrenocortical cancer

## 1. Introduction

Immune-checkpoint inhibitors (ICIs), including those targeting programmed cell death protein-1 (PD-1) receptors on T cells, are revolutionizing the treatment of solid organ and hematologic malignancies. Blocking the inhibitory signals of cytotoxic T cells, ICIs enable the upregulation of the antitumor immune response—harnessing the power of the lymphocytic system against cancer cells. Pembrolizumab is one such anti-PD-1 agent, with a growing list of approved indications including non-small-cell lung cancers, metastatic melanoma, and head and neck squamous cell cancer [1]. While ICIs hold immense potential in their direct clinical benefit, these inhibitors may also non-specifically activate the immune system, resulting in a subtype of side effects called immune-related adverse events (irAEs). Cutaneous toxicities are among the most common irAEs, occurring in 30–40% of patients treated with PD-1 inhibitors [2]. Maculopapular rash is the most commonly reported side effect, while vitiligo, lichenoid reactions, psoriasiform eruptions, and pruritus are also frequently observed [2,3]. Additionally, there have been well-established associations between ICIs with lichenoid dermatitis and autoimmune blistering disorders like bullous pemphigoid (BP) [4]. Far less frequently reported is lichen planus pemphigoides (LPP)—the unique crossover between lichen planus (LP) and BP.

LPP is a very rare autoimmune dermatosis which develops in the context of autoantibodies against BP180, a structural protein in the hemidesmosomes at the dermal–epidermal junction [4]. The estimated prevalence of this disorder is approximately 1 per million patients [4]. This subepidermal blistering disorder presents as pruritic, violaceous papules over distal extremities. It is then followed by bullous lesions on either lichenoid or normal skin [4,5]. Ultimately, the presence of autoantibodies at the dermal–epidermal junction, such as those against BP180 and

BP230, is critical to distinguishing the diagnosis of LPP from other similar disorders like bullous LP [5]. Immunofluorescence and enzyme-linked immunosorbent assay studies can provide valuable information, especially in the workup of patients presenting with atypical irAEs.

To date, relatively few cases of anti-PD-1 associated LPP have been reported. Herein, we report a patient who presented with LPP following pembrolizumab treatment for metastatic adrenocortical cancer (ACC). This is the first case, to our knowledge, that describes the presentation of LPP in the context of ACC—an exceedingly rare malignancy for which immunotherapy has only recently been indicated for. In addition to our case report of anti-PD-1-induced LPP, we also provide a summary of clinical, histopathologic, and immunofluorescence features of cases previously reported in the literature.

## 2. Case Report

A 46-year-old woman presented with a 3-week history of painful and pruritic blisters. Her pertinent past medical history includes an 8-month history of metastatic adrenocortical carcinoma, for which she was initially treated with six cycles of etoposide, doxorubicin, and cisplatin plus oral mitotane chemotherapy that was complicated by neuropathy. She was subsequently treated with four cycles of pembrolizumab, a PD-1 inhibitor. Hives and angioedema developed 11 days after the first dose of pembrolizumab, which resolved with oral antihistamines. She went on to receive three additional doses; however, pembrolizumab was discontinued after the fourth dose, as her metastatic disease had progressed. One week after her final infusion of pembrolizumab, the patient simultaneously developed tense bulla and pruritic plaques of the trunk and extremities. The physical exam was notable for diffuse lichenoid plaques and tense bullae on the chest, abdomen, and extremities with the greatest disease burden affecting the lower extremities (Figure 1). The exam was negative for oral mucosal findings like Wickham striae. A biopsy taken from the right thigh demonstrated subepidermal separation with a patchy dermal lymphocytic infiltrate, basilar dyskeratosis, necrosis, and presence of rare eosinophils (Figure 2A). Direct immunofluorescence (DIF) exhibited linear deposition of IgG and C3 along the basement-membrane zone (Figure 2B). The collective clinical, histopathologic, and immunofluorescence findings were consistent with a diagnosis of LPP. The patient was treated with topical and oral steroids, and the condition improved over a 3-month follow-up period without sequelae.

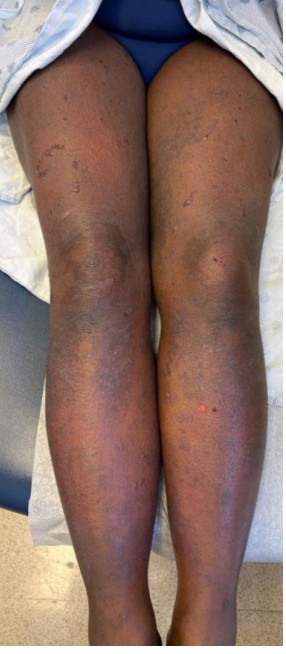

**Figure 1.** Clinical characteristics of patient included tense bulla and pruritic plaques disproportionately affecting the lower extremities. Physical exam was also notable for lichenoid plaques and tense bullae on the chest, abdomen, and extremities.

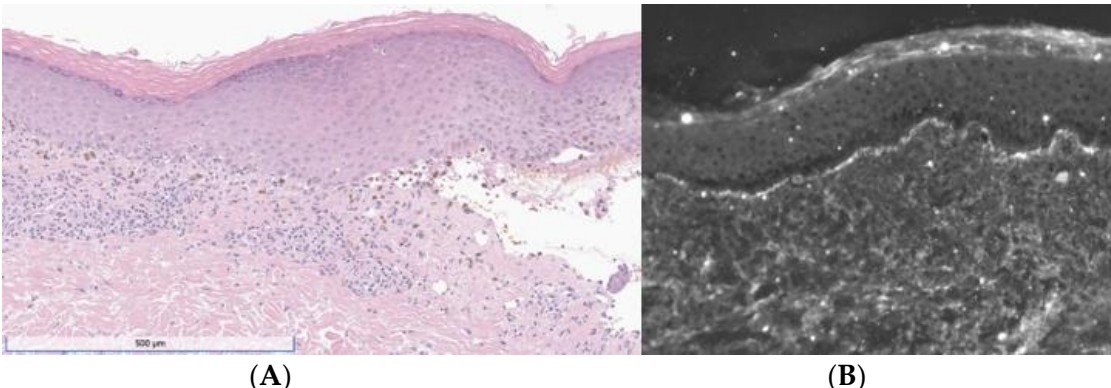

**(A)** **(B)**

**Figure 2.** (**A**) H&E showing band-like lymphocytic infiltrate with vacuolization of the basal layer, necrotic keratinocytes, pigment incontinence, and a subepidermal split. (**B**) DIF was positive for linear deposition of IgG (depicted) and C3.

## 3. Discussion

LPP, an uncommon autoimmune blistering disorder displaying characteristics of both LP and BP, is a rare manifestation of anti-PD-1 cutaneous toxicity. The diagnosis requires careful clinical correlation with histopathologic features, such as the presence of autoantibodies at the dermal–epidermal junction. Given the rarity of this irAE, we present our case report in conjunction with a summary of the literature on the topic. A comprehensive review of English-language medical literature was conducted using PubMed, identifying articles with terms including "lichen planus pemphigoides", "LPP", "PD-1 inhibitor", "pembrolizumab", "nivolumab", "tislelizumab", "PD-L1 inhibitor", "atezolizumab", or "durvalumab". A total of 20 cases, including this one, are summarized from reports published between April 2017 and April 2023.

Clinical features—such as patient demographics, immunotherapy course, and LPP presentation—are summarized in Tables 1 and 2. The mean onset age of LPP was 65.2 years, ranging between 12 and 84 years old. Among the 20 reported cases, there was a slight predominance of affected females (60%, 12/20) versus males (40%, 8/20)—mirroring the higher prevalence of LP and autoimmune blistering diseases among women [6,7]. Six (30%) patients identified as White, 4 (20%) identified as Black, 5 (25%) identified as Asian, and 5 (25%) did not report race. The majority (45%, 9/20) had underlying lung cancer, followed by melanoma (15%, 3/20) and urothelial cancer (15%, 3/20). Less common indications included breast cancer (5%, 1/20), Merkel cell carcinoma (5%, 1/20), renal cell carcinoma (5%, 1/20), hepatocellular carcinoma (5%, 1/20), and our case presentation of ACC (5%, 1/20). The majority of these malignancies were treated with pembrolizumab (55%, 11/20). Nivolumab (35%, 7/20) was another commonly used anti-PD-1 agent, with the remaining cases utilizing tislelizumab (5%, 1/20) or atezolizumab (5%, 1/20). The use of adjuvant therapy such as chemotherapy was noted in 20% (4/20) of patients, along with 10% (2/20) on targeted tyrosine kinase inhibitors like sitravatinib. Other medications utilized by patients were reported in five cases, with only one report of statin use and no use of angiotensin-converting enzyme (ACE) inhibitors. This is relevant given that LPP has been associated with ACE inhibitors, antituberculosis medications, and statins [4].

**Table 1.** Patient demographics and clinical characteristics of LPP presentation.

| Cases | Age | Sex | Race | Primary Disease | Cutaneous Presentation | Site | Mucosal | Nail | Therapy | Adjuvant Therapy |
|---|---|---|---|---|---|---|---|---|---|---|
| 1 (Wat et al., 2022) [8] | 80 | F | Black | Stage IV lung adenocarcinoma and cerebral metastases (cT3cN3cM1) | Erythematous-to-violaceous flat-topped papules, widespread erythema with tense and eroded blisters with some in annular configuration around central blister. | Upper and lower extremities, trunk, buttock, back | No | No | Pembrolizumab | Not reported |
| 2 (Wat et al., 2022) [8] | 77 | M | White | Stage IIIA non-small-cell lung cancer | Grade 3 mucositis, edematous pink papules and plaques with central erosions and crusting. | Upper and lower extremities | Yes | No | Pembrolizumab | Not reported |
| 3 (Wat et al., 2022) [8] | 63 | F | White | Metastatic breast cancer | Erythematous papules and plaques, flaccid bullae, and papular rash. | Upper and lower extremities, trunk | No | No | Pembrolizumab | Not reported |
| 4 (Kwon et al., 2020) [9] | 65 | F | White | Metastatic Merkel cell carcinoma | Erythematous-to-violaceous, lichenified, and hypertrophic papules and plaques. | Upper and lower extremities, trunk, back | No | No | Pembrolizumab | Not reported |
| 5 (Okada et al., 2020) [10] | 76 | M | Not reported | Advanced urothelial carcinoma | Pruritic erythematous patches and plaques, Wickham striae. Tense blisters additionally presented 13 wk later. | Lower extremity | Yes (reticulated white patches) | No | Pembrolizumab | Not reported |
| 6 (Sugawara et al., 2020) [11] | 72 | F | Asian | Stage IV lung cancer | Pruritic erythematous-to-violaceous papules and plaques. Tense blisters and edematous erythema presented 7 wk later. | Upper and lower extremities, trunk | No | No | Pembrolizumab | Not reported |
| 7 (Senoo et al., 2020) [12] | 76 | F | Asian | Stage IVA lung squamous cell carcinoma | Pruritic violaceous papules and plaques with Wickham's striae, vesicle eruption, and white reticular lesion. | Upper and lower extremities, back | Yes (reticulated white patches) | No | Atezolizumab | Not reported |

**Table 1.** *Cont.*

| Cases | Age | Sex | Race | Primary Disease | Cutaneous Presentation | Site | Mucosal | Nail | Therapy | Adjuvant Therapy |
|---|---|---|---|---|---|---|---|---|---|---|
| 8 (Schmidgen et al., 2017) [13] | 64 | M | White | Stage IV melanoma | Pruritic erythematous papules and plaques with central vesicles. Blisters on lichenoid plaques and on unaffected skin presented 26 wk later. | Upper and lower extremities, trunk | Yes (reticulated white patches) | No | Pembrolizumab | Standard chemotherapy as part of MK-3475-002 Phase II trial |
| 9 (Sato et al., 2019) [14] | 57 | M | Asian | Stage IVB lung squamous cell carcinoma | Erythematous lesions, papules and vesicles. | Upper and lower extremities, trunk | No | No | Nivolumab | Docetaxel and ramucirumab |
| 10 (Kerkemeyer et al., 2020) [15] | 75 | F | Not reported | Stage IV lung adenocarcinoma | Pruritic violaceous papules and plaques with tense bullae and Wickham's striae. | Upper and lower extremities, back | Yes (erythema and erosions) | No | Tislelizumab | Sitravatinib |
| 11 (Strickley et al., 2019) [16] | 87 | F | Not reported | Stage IV non–small-cell lung cancer | Erythematous crusted erosions and flaccid bullae. | Upper and lower extremities, trunk, back | No | No | Nivolumab | Not reported |
| 12 (Shah et al., 2022) [17] | 58 | F | Not reported | Renal cell carcinoma | Pruritic, erythematous, verrucous-like plaques and tense bullae. | Upper extremity | No | No | Nivolumab | Not reported |
| 13 (Liu et al., 2023) [18] | 60 | M | Black | Transitional cell carcinoma of the bladder and hepatocellular carcinoma | Erythematous to dusky cyanotic patches and targetoid plaques. Reemerged 3 wk later along with flaccid blisters and exfoliative skin with erythema; associated with positive Nikolsky sign. | Upper and lower extremities, back, scalp | Yes | No | Nivolumab | Not reported |
| 14 (Ee et al., 2022) [19] | 84 | M | Asian | Stage IV non–small-cell lung cancer | Pruritic, purpuric papules and blisters. | Upper and lower extremities, buttocks | No | No | Pembrolizumab | Not reported |

| Cases | Age | Sex | Race | Primary Disease | Cutaneous Presentation | Site | Mucosal | Nail | Therapy | Adjuvant Therapy |
|---|---|---|---|---|---|---|---|---|---|---|
| 15 (Mueller et al., 2022) [20] | 12 | M | Black | Stage IIIC spitzoid melanoma | Lichenoid papules and plaques with vesicles and bullae on background erythema. | Lower extremity | No | No | Nivolumab | Not reported |
| 16 (Yoshida et al., 2021) [21] | 70 | F | Not reported | Malignant melanoma | Papules and erythematous plaques with bullae. | Upper and lower extremities | No | No | Pembrolizumab | Not reported |
| 17 (Boyle et al., 2022) [22] | 66 | F | Asian | Stage IV urothelial cancer | Erythematous-to-violaceous papules and plaques with central erosions, flaccid bullae, and erosive mucositis. | Upper and lower extremities | Yes (reticulated white patches, hemorrhagic crust, and ulcerations) | Yes | Nivolumab | Sitravatinib |
| 18 (Boyle et al., 2022) [22] | 59 | M | White | Stage IV hepatocellular carcinoma | Erythematous-to-violaceous hyperkeratotic papules, crusted plaques with tense and flaccid bullae. | Lower extremity | No | No | Nivolumab | Not reported |
| 19 (Boyle et al., 2022) [22] | 57 | F | White | Stage IV non-small-cell lung cancer | Violaceous papules and plaques with central hyperkeratosis, small crusted papules. | Upper and lower extremities, trunk | Yes (hemorrhagic crust) | Yes | Pembrolizumab | Carboplatin and Paclitaxel |
| 20 (Our Case) | 46 | F | Black | Stage IV adrenocortical carcinoma | Diffuse lichenoid plaques, tense bullae. | Upper and lower extremities, trunk | No | No | Pembrolizumab | Mitotane |

**Table 2.** Temporal characterization of LPP onset and clinical management.

| Cases | Time to LPP from ICI Onset | Other Associated AEs | ICI Discontinued? | Did LPP Develop after Cessation of ICI? | Treatment |
|---|---|---|---|---|---|
| 1 (Wat et al., 2022) [8] | 45 wk | Lichen planus diagnosis 2 wk prior to LPP, desquamation of 44% of total body surface area | Yes | No | IV corticosteroids, followed by prednisone |
| 2 (Wat et al., 2022) [8] | 26 wk | Low-grade pneumonitis, lymphocytic enterocolitis | Yes | No | Prednisone |
| 3 (Wat et al., 2022) [8] | 17 wk | Bullous tinea diagnosis 3 wk prior to LPP | Temporarily (resumed after eruption control) | No | Doxycycline and nicotinamide |
| 4 (Kwon et al., 2020) [9] | 1 wk | None | Yes | No | Oral prednisone, clobetasol |
| 5 (Okada et al., 2020) [10] | 12 wk | Not reported | Yes (due to disease progression) | Yes (13 wk later) | Oral prednisone |
| 6 (Sugawara et al., 2020) [11] | 20 wk | Not reported | Yes (due to disease progression) | Yes (7 wk later) | Oral prednisone |
| 7 (Senoo et al., 2020) [12] | 17 wk | Not reported | Yes (due to disease progression) | No | Topical corticosteroids |
| 8 (Schmidgen et al., 2017) [13] | 79 wk | CTCAE Grade 2 BP-like reaction, thrombocytopenia, heart failure | Yes (due to BP-like reaction 26 wk before LPP) | Yes (52 wk later) | Dapsone |
| 9 (Sato et al., 2019) [14] | 26 wk | Hand–foot syndrome | Yes (due to disease progression) | Yes (13 wk later) | Oral prednisolone with doxycycline |
| 10 (Kerkemeyer et al., 2020) [15] | 6 wk | Not reported | Not reported | Not reported | Prednisolone, topical betamethasone |
| 11 (Strickley et al., 2019) [16] | 13 wk | Not reported | Yes | No | Oral prednisone |
| 12 (Shah et al., 2022) [17] | 17 wk | Not reported | Yes | No | Topical clobetasol cream, systemic corticosteroids, and oral pregabalin |
| 13 (Liu et al., 2023) [18] | 33 wk | Not reported | Yes | No | Prednisone |
| 14 (Ee et al., 2022) [19] | 11 wk | Not reported | No | N/A | Topical clobetasol |

**Table 2.** *Cont.*

| Cases | Time to LPP from ICI Onset | Other Associated AEs | ICI Discontinued? | Did LPP Develop after Cessation of ICI? | Treatment |
|---|---|---|---|---|---|
| 15 (Mueller et al., 2022) [20] | 10 wk | Severe skin pain, joint swelling, and rapidly spreading vesiculobullous lesions | Yes | No | Topical betamethasone dipropionate and IV methylprednisolone transitioned to oral prednisolone and methotrexate |
| 16 (Yoshida et al., 2021) [21] | 21 wk | Not reported | Yes | No | Oral prednisolone |
| 17 (Boyle et al., 2022) [22] | 4 wk | Not reported | Yes | No | Oral prednisone and topical clobetasol, followed by rituximab |
| 18 (Boyle et al., 2022) [22] | 48 wk | Not reported | Yes | No | Topical clobetasol |
| 19 (Boyle et al., 2022) [22] | 6 wk | Not reported | No (eventually discontinued 5 cycles later due to disease progression) | N/A | Oral prednisone and topical clobetasol |
| 20 (Our Case) | 13 wk | Hives, angioedema | Yes (due to disease progression) | Yes (1 wk later) | Oral prednisone and topical tetracaine–adrenaline–cocaine |

In our case and review of the literature, the presentation of LPP following immunotherapy ubiquitously included the formation of erythematous papules and plaques on the extremities. Associated pruritus was observed in 35% (7/20) of cases, and a delayed blister presentation was reported in 4 cases, ranging from 3–26 weeks after the first observation of lichenoid plaques. In addition to the upper and lower extremities, another common site of LPP was the trunk (40%, 9/20), back (25%, 5/20), and scalp (5%, 1/20). Mucosal involvement, most often presenting as reticulated white patches, was observed in 45% (9/20) of patients. Nail-bed involvement, noted in 10% (2/20) of cases, was far less common.

The average time to LPP onset from ICI commencement was 21.25 weeks, with a range of 1–79 weeks. Our patient's presentation of LPP arose one week after the final dose of pembrolizumab, corresponding to the 12-week mark since initiation. Other associated irAEs were noted in 35% (7/20) of patients, ranging from dermatologic manifestations like hives, angioedema, joint swelling, bullous tinea, and hand–foot syndrome to more systemic irAEs like pneumonitis and heart failure. For 6 (30%) patients, ICI was discontinued prior to the onset of LPP due to either disease progression (5/20, 25%) or earlier irAE (1/20, 5%). The average time between ICI cessation and LPP diagnosis was 17.2 weeks, ranging between 1–52 weeks. This latency period, also observed in our patient, is of particular note as cutaneous toxicities can present even after cessation of immunotherapy. Among patients who were actively on immunotherapy when LPP presented (70%, 14/20), all but two (10%) had subsequent discontinuation of ICI. The majority of patients were treated with either oral or topical corticosteroids (90%, 18/20), with the remaining cases (10%, 2/20) managed with antibiotics. In the case of severe cutaneous AEs, current guidelines recommend immunotherapy withdrawal and treatment with systemic steroids [23].

Histopathologic and immunofluorescence findings are highlighted in Table 3. Band-like lichenoid dermatitis was identified in all cases, often accompanied by basal-layer vacuolization (55%, 11/20), hyperkeratosis (55%, 11/20), lymphocytic infiltrate at the dermal–epidermal junction (45%, 9/20), and dyskeratotic keratinocytes (45%, 9/20). Subepidermal blisters were reported in 15 cases (75%) and eosinophilic infiltrate was observed in 10 cases (50%). DIF revealed linear C3 or IgG deposition for 95% of patients (18/19), with 68% (13/19) presenting with both C3 and IgG deposition. Four cases (21%) reported only a positive C3 deposition, and one case (5%) was positive for a linear IgG deposition at the dermal–epidermal junction. One case had negative DIF findings, but indirect immunofluorescence (IIF) demonstrated a weak linear IgG deposition at the basement-membrane zone on monkey esophagus. Another case revealed a DIF finding of fibrinogen along the basement-membrane zone. This finding, while not frequently reported in the reviewed cases of LPP, has been independently associated with both LP and BP [24,25].

IIF was not performed for the majority of cases (75%, 15/20), but among those that did, 4 cases (80%) were positive for linear IgG on salt-split skin. Serum enzyme-linked immunosorbent assay (ELISA) was performed in 70% of cases (14/20), while 10% (2/20) did not conduct this assay and 20% (4/20) did not report. Among those that performed ELISA, 86% (12/14) were positive for elevated anti-BP180 titers. Anti-BP230 titer was also elevated in one of these cases (8%, 1/12), and negative in 2 (16%, 2/12). The remainder of cases with positive BP180 antibodies (75%, 9/12) did not report serum anti-BP230 titers. Two patients (10%, 2/20) revealed negative anti-BP180 and negative anti-BP230 titers. In one of these cases, however, an immunoblotting revealed IgG antibodies specific to C-terminus of BP180 [21]. The second patient with negative BP180 and BP230 antibodies had a unique presentation of follicular immunobullous dermatosis, as opposed to the subepidermal bullae at the nonfollicular dermal–epidermal junction observed in the other reported LPP cases [18]. The authors proposed the role of a novel antibody mediating the presentation of LPP at the hair follicle—suggesting a unique target at the perifollicular basement membrane that triggered a lichenoid reaction [18].

Table 3. Histopathologic and immunofluorescence features of LPP cases.

| Cases | Biopsy Site | Histology | DIF | IIF | ELISA BP190/BP230 |
|---|---|---|---|---|---|
| 1 (Wat et al., 2022) [8] | Left thigh | Acanthotic epidermis, hyperkeratosis, hypergranulosis, lichenoid infiltrate and basal layer vacuolization. Subepidermal blister formation. | Linear C3 and IgG at BMZ | Not performed | Positive BP180, Negative BP230 |
| 2 (Wat et al., 2022) [8] | Right forearm, buccal mucosal | Cutaneous: Acanthotic epidermis with a band-like lichenoid infiltrate and basal-layer vacuolization and necrotic keratinocytes. <br><br> Oral mucosal: focal basal cell layer degeneration, lichenoid mucositis, acanthosis, and perivascular lymphocytic and plasma cell infiltrate. | Cutaneous: Linear C3 and IgG at BMZ. <br><br> Oral mucosa: Linear C3 at BMZ. | Not performed | Positive BP180, Negative BP230 |
| 3 (Wat et al., 2022) [8] | Right palm, left leg | Palm: Lichenoid tissue reaction with lymphocytic infiltrate at BMZ, dyskeratosis, perivascular dermal inflammation. <br><br> Leg: Subepidermal blister with a necrotic roof, focal interface reaction with lymphocytes. | Not performed | Not performed | Positive BP180 |
| 4 (Kwon et al., 2020) [9] | Not reported | Lichenoid and vacuolar epidermal interface alteration with dyskeratotic keratinocytes and eosinophils. | Linear C3 and IgG at BMZ | Linear IgG on salt-split skin | Positive BP180 |
| 5 (Okada et al., 2020) [10] | Lower extremity | Orthokeratosis, irregular acanthosis, and band-like inflammatory infiltrates in upper dermis. Rete ridge base with accentuated vacuolar alteration and lymphocytic infiltrates. Later biopsies showed a subepidermal blister with eosinophilic infiltrate in upper dermis. | Linear C3 and IgG at BMZ | Not performed | Positive BP180 |
| 6 (Sugawara et al., 2020) [11] | Lower extremity | Orthokeratosis with irregular acanthosis and hypergranulosis. Vacuolar alteration of basal layer and dermal lymphocyte infiltration. Later biopsies showed subepidermal blister with eosinophilic infiltration. | Linear C3 and IgG at BMZ | Not performed | Positive BP180 |
| 7 (Senoo et al., 2020) [12] | Thigh | Band-like lymphocytic infiltrate in upper dermis with vacuolar degeneration of BMZ. Hypergranulosis, subepidermal blister with mixed infiltrate of eosinophils and lymphocytes. | Linear C3 at BMZ, no IgG | Negative | Positive BP180 |

**Table 3.** *Cont.*

| Cases | Biopsy Site | Histology | DIF | IIF | ELISA BP190/BP230 |
|---|---|---|---|---|---|
| 8 (Schmidgen et al., 2017) [13] | Sacral region | Lichenoid dermatitis with band-like lymphocytic infiltrate at BMZ. Orthohyperkeratosis, hypergranulosis, cytoid bodies. Later biopsies showed subepidermal blistering and eosinophil infiltrate. | Linear C3 at BMZ | IgG binding to epidermal side on salt-split skin | Positive BP180 |
| 9 (Sato et al., 2019) [14] | Not reported | Band-like lymphocytic infiltrate with prominent eosinophils at BMZ. Vacuolar degeneration with apoptotic keratinocytes. | IgG deposition at BMZ | Not reported | Positive BP180 |
| 10 (Kerkemeyer et al., 2020) [15] | Not reported | Acanthosis, parakeratosis, and elongation of rete ridges with basal cell degeneration. Subepidermal bulla with a small number of eosinophils, and perivascular mixed inflammatory infiltrate with occasional eosinophils of the upper dermis. | Linear C3 and IgG at BMZ | Not reported | Not reported |
| 11 (Strickley et al., 2019) [16] | Left thigh | Lichenoid dermatitis followed by subepidermal bullous lichenoid eruption with eosinophils. | Negative | Weak linear IgG at BMZ on monkey esophagus | Not reported |
| 12 (Shah et al., 2022) [17] | Not reported | Acanthotic epidermis with a band-like inflammatory infiltrate and basal layer vacuolization. Subepidermal blister formation with sparse dermal perivascular infiltrate. | Linear C3 at BMZ | Not performed | Not reported |
| 13 (Liu et al., 2023) [18] | Right parietal scalp | Lichenoid dermatitis with perifollicular clefting. Orthokeratosis, hypergranulosis, sawtooth-like change in BMZ, dyskeratotic keratinocytes, and colloid bodies. | Linear C3 and IgG at BMZ and in a perifollicular distribution | Not performed | Negative BP180 and negative BP230 |
| 14 (Ee et al., 2022) [19] | Right shin | Lichenoid infiltrate of lymphocytes, histiocytes, and eosinophils. Basal vacuolar alteration and subepidermal clefting. | Linear C3 and IgG at BMZ | Not reported | Not reported |
| 15 (Mueller et al., 2022) [20] | Right thigh | Subepidermal separation with basal necrotic keratinocytes and sparse lichenoid lymphocytic infiltrate. | Linear C3 at BMZ, no IgG | Not performed | Positive BP180 |

**Table 3.** *Cont.*

| Cases | Biopsy Site | Histology | DIF | IIF | ELISA BP190/BP230 |
|---|---|---|---|---|---|
| 16 (Yoshida et al., 2021) [21] | Thigh | Biopsy 1: Lichenoid dermatitis with lymphocytic infiltration at BMZ, hyperkeratosis, sawtooth-like acanthosis, cytoid bodies, hypergranulosis. Biopsy 2: Subepidermal blister and dermal lymphocytic infiltration. | Linear C3 and IgG at BMZ | IgG binding to epidermal side on salt-split skin | Negative BP180 and negative BP230 |
| 17 (Boyle et al., 2022) [22] | Left foot | Lichenoid dermatitis with dyskeratotic keratinocytes at lower epidermis. Orthokeratosis, hypergranulosis, and subepidermal bullae. | Linear IgG and C3, cytoid bodies, and retiform fibrinogen at BMZ | Not performed | Positive BP180, Positive BP230 |
| 18 (Boyle et al., 2022) [22] | Left leg | Acanthotic epidermis, orthokeratosis, and basal-layer vacuolization with dyskeratotic keratinocytes. Superficial-to-deep perivascular and interstitial inflammatory infiltrate with many eosinophils in dermis. | Linear IgG and C3 at BMZ, cytoid bodies in papillary dermis | Not performed | Not performed |
| 19 (Boyle et al., 2022) [22] | Right palm | Lichenoid dermatitis with dyskeratotic keratinocytes at basal epidermis. Orthokeratosis, focal paragranulosis, hypergranulosis, and neutrophils in stratum corneum. | Linear C3 and IgG at BMZ | Linear IgG on monkey esophagus with epidermal pattern on salt-split skin. Negative on rat bladder | Positive BP180, Negative BP230 |
| 20 (Our Case) | Right thigh | Band-like lymphocytic infiltrate with vacuolization of the basal layer, necrotic keratinocytes, pigment incontinence, and a subepidermal split. | Linear C3 and IgG at BMZ | Not performed | Not performed |

The mechanism by which PD-1 inhibitors induce LPP is not fully understood. Clinically, the majority of LPP cases follow a timeline where lichenoid skin lesions precede the formation of tense blisters. Thus, it has been suggested that lichenoid inflammation itself may promote an autoimmune response against subepidermal proteins [26,27]. One hypothesis is that the increased cytotoxic CD8+ T cells from PD-1 inhibition renders extensive apoptosis of the basal epidermis, thereby exposing various antigens of the dermal–epidermal junction to autoreactive T cells [4]. This antigen presentation may enable the formation of autoantibodies, leading to the classic presentation of subepidermal blisters seen in LPP [27]. Details surrounding this hypothesized mechanism remain elusive given the low frequency of LPP.

To our knowledge, this is the first reported case of LPP associated with pembrolizumab treatment for metastatic adrenocortical cancer (ACC). ACC is a very aggressive and rare endocrine malignancy with an estimated incidence of 0.5 to 2 cases per million individuals [28,29]. Given the poor prognosis of ACC, efforts are being made to utilize immunotherapy as salvage therapy. Our patient was also started on pembrolizumab upon the failure of first-line mitotane therapy. Unfortunately, the anti-PD-1 salvage therapy was discontinued as the metastatic disease progressed.

Interestingly, multiple studies have shown an association between cutaneous toxicities and superior clinical outcomes. The development of dermatological AEs has been correlated with increased tumor response rate, progression-free survival, and overall survival [26,30]. It is hypothesized that the presentation of irAEs can serve as a surrogate marker for ICI efficacy [26]. However, the majority of these studies have focused on more common cutaneous AEs such as maculopapular rash, vitiligo, and hypopigmentation. There is currently no assessment of the relationship between LPP development and treatment outcome.

Since the first FDA approval of PD-1 inhibitors in 2014, the clinical development of such agents has gained much traction. There are currently 5 PD-1 inhibitors and 3 PD-L1 inhibitors on the market, with many more under ongoing investigation. Additionally, the indications for these immunotherapy drugs continue to expand as clinical trials include broader sets of neoplasms like ACC. For instance, just over the past decade, the indications for pembrolizumab have grown from advanced melanoma to numerous solid tumors including lung cancer, urothelial cancer, cervical cancer, esophageal cancer, renal cell carcinoma, and triple-negative breast cancer, to name a few (Figure 3). Additionally, the efficacy of pembrolizumab extends to hematologic malignancies and tissue-agnostic tumors based on specific molecular phenotypes. As the number and indications of anti-PD-1 agents continue to grow in the foreseeable future, it becomes more likely to see rare immune-related toxicities like LPP. As of June 2020, for instance, there were only 10 reported cases of anti-PD-1-induced LPP compared to the 20 patients presented in this review. The steep rise in reported cutaneous irAEs further underscores the careful surveillance and awareness needed as immunotherapy use becomes more widespread.

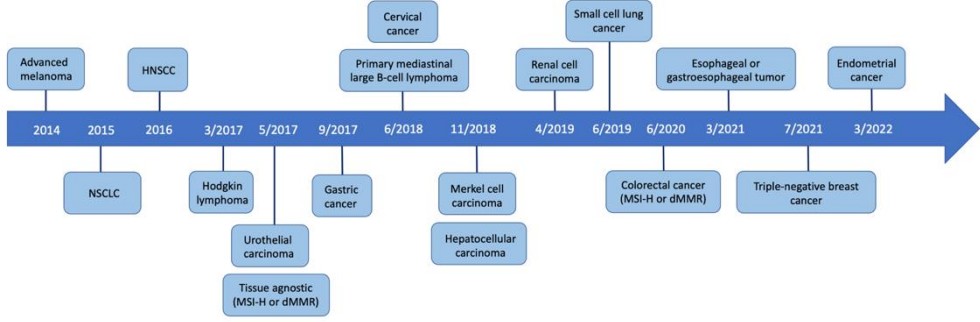

**Figure 3.** Timeline of FDA-approved indications for pembrolizumab.

## 4. Conclusions

In summary, we report the first known case of LPP following anti-PD-1 therapy for metastatic ACC. While LPP is rare independently, its presentation with an orphan

malignancy like ACC makes this case all the more unique. We also provide a summary of clinical, histopathologic, and immunofluorescence findings of 20 immunotherapy-induced LPP cases in the literature. Clinical presentation includes erythematous papules and plaques favoring the extremities and trunk. The delayed presentation of bullae after plaque formation was observed, in addition to mucosal involvement in about half the cases. Histopathology of LPP presented with a band-like lichenoid dermatitis with subepidermal bullae, often accompanied by vacuolization or hyperkeratosis. DIF revealed linear C3 or IgG, while IIF, when performed, often demonstrated linear IgG along the basement-membrane zone. Serum anti-BP180 levels were also elevated in the majority of cases. Recognizing that LPP is within the spectrum of cutaneous AEs is important as the early recognition and prompt treatment of such irAEs is critical for patient safety. This case highlights key features of LPP that can help broaden our understanding of the growing number of cutaneous toxicities in the context of PD-1 inhibitors.

**Author Contributions:** Conceptualization, V.M., M.C.M. and J.C.S.; writing—original draft preparation, V.M.; writing—review and editing, M.C.M. and J.C.S. All authors have read and agreed to the published version of the manuscript.

**Funding:** This research received no external funding.

**Institutional Review Board Statement:** Ethical review and approval were waived for this study due to deidentified patient information, informed consent, and publication of a case report as opposed to original investigation.

**Informed Consent Statement:** Written informed consent has been obtained from the patient to publish this paper.

**Data Availability Statement:** No new data were created or analyzed in this study. Data sharing is not applicable to this article.

**Conflicts of Interest:** The authors declare no conflict of interest.

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
