# Peer review of "Pembrolizumab-Induced Lichen Planus Pemphigoides in a Patient with Metastatic Adrenocortical Cancer: A Case Report and Literature Review"

_dermatopathology, doi:10.3390/dermatopathology10030033_

Round 1

Reviewer 1 Report

This is a good paper with very thorough review of the literature providing a very compact reference for anyone looking to know more about checkpoint inhibitor induced LPP. 

I do have one big question about the direct immunofluorescence analysis. What was the interpretation of fibrinogen along the basement membrane zone? Are you able to comment on this for this specific case as well as the cases presented in the literature? Typically, shaggy fibrinogen can be noted in lichenoid reactions such as lichen planus and I wonder how frequent this finding was in the cases reviewed. Only one case mentions it in table 3. I think this deserves some commentary in this paper.

Author Response

Response to Reviewer 1 Comments

Point 1:  I do have one big question about the direct immunofluorescence analysis. What was the interpretation of fibrinogen along the basement membrane zone? Are you able to comment on this for this specific case as well as the cases presented in the literature? Typically, shaggy fibrinogen can be noted in lichenoid reactions such as lichen planus and I wonder how frequent this finding was in the cases reviewed. Only one case mentions it in table 3. I think this deserves some commentary in this paper.

Response 1: This is a great point. Fibrinogen at the basement membrane zone was only reported in one of the reviewed LPP cases. However, this finding has been commonly found in oral LP and has also occasionally been associated with BP. Its presence in LPP remains a question that would require additional follow up. Lines 183-185 address this.

Reviewer 2 Report

This is a case report of Lichen Planus Pemphigoides (LPP) developing after pembrolizumab therapy for adrenocortical cancer (ACC). The authors performed a thorough literature review of LPP cases developing secondary to anti-PD-1 therapy for various malignancies.

This article is rather well-written and the review carefully performed. I have however some suggestions for improvement and some questions/comments:

- line 6: the affiliation of the authors seems incomplete – please check

- line 40: collagen XVII is the same as BP180. I suggest to use consistently the same term (BP180) for the sake of clarity.

- lines 44-45: BP230 and BP180 are antigens, not autoantibodies! Please rephrase

- line 46: the presence of autoantibodies to BP230 and/or BP180 is important to distinguish LPP from Lichen Planus (LP) but not from BP; these autoantibodies are present (by definition) also in BP.  Please delete ‘BP’ from this sentence

- Case report: this could be more precise. Line 57: the exact age of the patient could be provided. Did the patient have typical LP papules/with Wickham striae? Did she develop LP/lichenoid lesions before the bullae? How long before? What was the outcome following treatment? Length of follow-up?...

- lines 98-99: autoimmune diseases are indeed more frequent in women, but this concerns mainly connective tissue diseases. I don’t think BP or LP show a female predominance – if relevant data exist, a reference should be provided here

- Tables: the articles presented in the tables are not included in the list of references! They should be included, and the numbering of the citations in the text corrected accordingly.

- For the sake of completeness, the outcome of the published LPP cases after treatment could be provided (eg as a last column ‘Outcome’ after ‘Treatment’ in Table 2). Similarly, the authors could review the outcome of the underlying malignancy in patients with LPP (so as to have a preliminary answer to the question raised in lines 191-192).

- lines 141-142: I am puzzled by the fact that only 40% of the reported LPP cases had histologically subepidermal blisters. This finding is essential for the diagnosis of LPP and the distinction from LP. Is there an explanation? Are we sure the remaining cases were indeed LPP?

- the authors repeatedly stress the fact that this is the ‘first case of LPP developing after anti-PD-1 therapy for adrenocortical carcinoma. In my view, this is not very relevant as LPP is due to the ICI, not to the underlying disease. Therefore I suggest to tone down the references to this association ,which seems fortuitous. Accordingly, the corresponding paragraph (lines 176-184) could be deleted or substantially shortened.

- line 142: 4/5 cases where IIF was performed were positive, therefore the corresponding percentage is 80% (not 20% - as below correctly stated that 86%: 12/14 cases with ELISA performed were positive).

- figure 2B: this could be given in color

- References: they are not presented in a uniform style: some have the first author et al, others have the full list of authors, some have journal abbreviations, others do not. Please read the ‘guidelines to authors’ of the journal and provide a correct presentation.

There are some typos that need correction (line 29: ‘…melanoma. and head….’, line 87: ‘…such as the presence autoantibodies at the dermal-epidermal junction’.). Table 3, p. 10, line 3: what is ‘paragranulosis’ ?

Author Response

- line 6: the affiliation of the authors seems incomplete – please check

Additional details were added, please see line 6.

- line 40: collagen XVII is the same as BP180. I suggest to use consistently the same term (BP180) for the sake of clarity.

Collagen XVII was replaced with BP180 at line 40.

- lines 44-45: BP230 and BP180 are antigens, not autoantibodies! Please rephrase

At line 46-48, this was rephrased to:“Ultimately, the presence of autoantibodies at the dermal-epidermal junction, such as those against BP180 and BP230, is critical to distinguishing the diagnosis of LPP from other similar disorders like bullous LP [5].”

- line 46: the presence of autoantibodies to BP230 and/or BP180 is important to distinguish LPP from Lichen Planus (LP) but not from BP; these autoantibodies are present (by definition) also in BP.  Please delete ‘BP’ from this sentence

BP was deleted from line 48.

- Case report: this could be more precise. Line 57: the exact age of the patient could be provided. Did the patient have typical LP papules/with Wickham striae? Did she develop LP/lichenoid lesions before the bullae? How long before? What was the outcome following treatment? Length of follow-up?...

The exact age was provided at line 59. The patient did not present with typical LP papules with Wickham striae. Additionally, this was atypical in that the lichenoid lesions presented simultaneously with bullae. The outcome was recovery without recurrence over a 3 month follow up. This has been included in revisions of the paragraph from lines 59-78.

- lines 98-99: autoimmune diseases are indeed more frequent in women, but this concerns mainly connective tissue diseases. I don’t think BP or LP show a female predominance – if relevant data exist, a reference should be provided here

Good point. The literature has shown a slight predominance in BP and LP. The relevant references were added at line 108.

- Tables: the articles presented in the tables are not included in the list of references! They should be included, and the numbering of the citations in the text corrected accordingly.

The citations have been updated accordingly.

- For the sake of completeness, the outcome of the published LPP cases after treatment could be provided (eg as a last column ‘Outcome’ after ‘Treatment’ in Table 2). Similarly, the authors could review the outcome of the underlying malignancy in patients with LPP (so as to have a preliminary answer to the question raised in lines 191-192).

This is an excellent point. I was also hoping to include outcome of both the LPP treatment and underlying malignancy. However, in the cases reviewed, only a few reported LPP treatment outcome – all of which were complete resolution as it appears that steroids are largely successful in treatment. Given the small number of reported cases, I decided that a column with so many empty data points may not be very helpful.

As for the underlying malignancy, most cases reviewed do not discuss the status of cancer treatment – focusing primarily on the LPP aspect of the presentation. It would be a very interesting question to explore how cutaneous manifestations like LPP correlate with immunotherapy success.

- lines 141-142: I am puzzled by the fact that only 40% of the reported LPP cases had histologically subepidermal blisters. This finding is essential for the diagnosis of LPP and the distinction from LP. Is there an explanation? Are we sure the remaining cases were indeed LPP?

Thank you for bringing this up. When searching for the term “subepidermal blister,” synonyms were not initially included in our query. Upon refinement, 75% of cases were found to report subepidermal blistering whereas the other 25% focused on the lichenoid histological changes. Line 165 incorporates this change.

- the authors repeatedly stress the fact that this is the ‘first case of LPP developing after anti-PD-1 therapy for adrenocortical carcinoma. In my view, this is not very relevant as LPP is due to the ICI, not to the underlying disease. Therefore I suggest to tone down the references to this association ,which seems fortuitous. Accordingly, the corresponding paragraph (lines 176-184) could be deleted or substantially shortened.

Agreed. This paragraph was shortened to remove information about the phase II trial.

- line 142: 4/5 cases where IIF was performed were positive, therefore the corresponding percentage is 80% (not 20% - as below correctly stated that 86%: 12/14 cases with ELISA performed were positive).

Thank you for pointing this out. Changes were made on line 187.

- figure 2B: this could be given in color

Unfortunately, we only have the black and white version of this.

- References: they are not presented in a uniform style: some have the first author et al, others have the full list of authors, some have journal abbreviations, others do not. Please read the ‘guidelines to authors’ of the journal and provide a correct presentation.

We have now updated the references to reflect MDPI Chicago style using RefWorks.